# Relationship between Village Chicken Availability and Dietary Diversity along a Rural–Urban Gradient

**DOI:** 10.3390/nu16132069

**Published:** 2024-06-28

**Authors:** Cresswell Mseleku, Michael Chimonyo, Rob Slotow, Mjabuliseni S. Ngidi

**Affiliations:** 1School of Agricultural, Earth and Environmental Sciences, https://ror.org/04qzfn040University of KwaZulu-Natal, Private Bag X01, Scottsville 3209, South Africa; 2Faculty of Science, Engineering and Agriculture, https://ror.org/0338xea48University of Venda, Private Bag X5050, Thohoyandou 0950, South Africa; 3School of Life Sciences, https://ror.org/04qzfn040University of KwaZulu-Natal, Private Bag X01, Scottsville 3201, Pietermaritzburg 3209, South Africa

**Keywords:** eggs, food variety score, food system, meat, nutritional security, offal

## Abstract

Dietary diversity is one of the fundamental factors of nutritional security and a proxy used to measure diet quality. The objective of this study was to investigate the relationship between village chicken availability and the dietary diversity of households along a rural–urban gradient. Face-to-face interviews were conducted using a structured questionnaire in rural (*n* = 100), peri-urban (*n* = 100), and urban (*n* = 100) areas of Pietermaritzburg uMgungundlovu District, KwaZulu-Natal, in South Africa. A positive relationship between distance from the city center and village chicken flock sizes (*p* < 0.001) was observed. Consumption of vegetables increased with an increase in distance from the city center (*p* < 0.01). A quadratic relationship was observed between distance from the city center and consumption of livestock-derived foods (LDFs) (*p* < 0.05). Consumption of LDFs increased with an increase in village chicken flock sizes (*p* < 0.05). Consumption of vegetables increased with an increase in village chicken flock sizes (*p* < 0.01). Food variety score (FVS) increased with an increase in distance from the city center (*p* < 0.05). Assessing the availability of village chickens across rural–urban gradients is a worthy opportunity to utilize to improve households’ dietary diversity and alleviate poverty. It can be concluded that expanding village flock sizes could enhance the dietary diversity of households.

## Introduction

1

Nutritional and health requirements are easily met when diets are diverse. Dietary diversity is one of the fundamental factors of nutritional security and a proxy used to measure diet quality [1,2]. Dietary diversity is poor in low- and middle-income countries (LMICs), and this poses a serious challenge especially in vulnerable groups due to their high nutritional needs [3]. In Africa, there are approximately one billion people who cannot afford a healthy diet [4]. In 2018, 25% of households in South Africa were food-insecure [5,6]. Approximately 80% of households in South Africa are involved in agriculture in order to secure additional food sources [6-9]. Village chickens form an integral part of nearly all rural households, many peri-urban households, and some urban households [5].

Rearing village chickens is one of the sustainable strategies that potentially improves the access of households to quality foods [10]. Rearing village chickens is convenient, mainly due to the low maintenance required and their adaptability, survivability, and scavenging activity [11]. Village chicken husbandry systems are inexpensive and require minimal land for production [12].

Although it has always been popular in rural communities, rearing village chickens is increasingly becoming the norm in urban and peri-urban environments [7]. Village chicken production at a household level has been rapidly recognized as one approach to improve the nutritional status of household members [11]. Understanding how village chicken rearing is linked to dietary diversity assists in designing sustainable programs that enhance the welfare of vulnerable household members [13]. Village chicken ownership is particularly important in resource-limited households that heavily rely on subsistence farming to compensate for their lack of purchasing power [14].

Improving dietary diversity is important to reduce all forms of malnutrition [3]. The diets of vulnerable households are commonly deficient in vitamin A and iron [15,16]. Households in LMICs are characterized by having less diverse diets, mainly defined by starchy staples with less protein sources such as meat and eggs [17]. In South Africa, a study that assessed micronutrient intakes in rural and urban areas revealed that iron and vitamin B levels were low in diets [18]. This is due to the poor quantity, quality, and diversity of diets [14]. Meat, eggs, and offal from village chickens contain high-quality protein and adequate amounts and proportions of zinc, iron, and vitamins A and B that promote optimal development [19]. Consumption of offal is popular in many resource-poor communities [20]. The contribution of village chickens is, however, poorly investigated as part of the food system [8]. There is little, if any, available information on the contribution of village chickens to household dietary diversity across a rural–urban gradient, where rural–urban gradient refers to the distance from the city center. The objective of this study was to investigate the relationship between village chicken availability and the dietary diversity of households along a rural–urban gradient. The relationship between village chicken flock sizes and dietary diversity was hypothesized to be linear.

## Materials and Methods

2

### Ethical Clearance

2.1

The study participants’ rights, religions, culture, and dignity were not violated. The study participants were assured that their confidential information would not be disclosed, and they had a right to stop the interview whenever they felt uncomfortable. The experimental procedures were performed according to the ethical guidelines specified by the Certification of Authorization to Experiment on Living Humans provided by the Social Sciences–Humanities and Social Sciences Research Ethics Committee, Protocol Reference No. HSSREC/00005927/2023.

### Study Site and Design

2.2

The study was conducted in Pietermaritzburg, Msunduzi Municipality, in uM-gungundlovu District (29.6006° S, 30.3794° E) in the KwaZulu-Natal province of South Africa. Twelve sites were selected to create a rural–urban gradient [17]. The sites representing the urban settlements lay within 10 km from the city center, while the peri-urban residential areas were located between 10 and 40 km from Pietermaritzburg city center. The rural settlements were between 40 and 70 km from the city center. The sites representing the rural environment were Impendle, Nkumba, Swayimana, and KwaHlwemini, while Gezubuso, Taylors Halt, Maqongqo, and Ezibananeni were regarded as peri-urban settlements, and Imbali, Edendale, Sobantu, and Eastwood represented urban areas. Rural areas have agricultural potential but are mostly suitable for animal husbandry. Poor secondary roads (gravel) that are not conducive for transport, especially in rainy conditions, are common. Most of the households use pit toilets, and the areas are distant from the city center.

Peri-urban areas have a high unemployment rate, and farming is the primary economic activity for many households [18].

The sites were selected based on the availability of village chickens and location from the city center and to characterize a rural–urban gradient. A transect approach was used to guide data collection. This approach increases the probability of including communities that may be excluded in most sampling approaches. Transects are lines that show the area of interest where sampling is focused [17]. Two transects along the southwest and north-east of Pietermaritzburg, uMgungundlovu District, cutting through the city were used to characterize a rural–urban gradient (Figure 1). Each transect was 70 km long and divided into rural, peri-urban, and urban settlements. A similar approach to that of Chagomoka et al. [17] was used to establish the distances.

### Selection of Participants and Data Collection

2.3

Pre-tested topic guides were used to conduct focus group discussions (FGDs). This enabled the researchers to obtain perspectives connected to the subjects studied. The FGDs were held with livestock association members and local leaders. The livestock association members and local leaders both comprised men and women. The FGDs were held at the homesteads of local leaders and community halls and conducted by the researchers. Participants of the FGDs were separated into small groups. Each group consisted of 5–9 participants. Participants were grouped according to their age and gender to ensure that they were comfortable to share information. Discussions were organized to acquire views on village chicken ownership and the dietary diversity of households. The FGDs lasted for about an hour and there was no food item adaptation applied during the discussions. The age of the selected participants was 18 years or above. The data gathered from the discussions were used to construct a structured questionnaire. The FGDs, questionnaires, and 24-hour dietary recalls were conducted in September 2023. The focus group discussions and questionnaires were written in IsiZulu, the local language, in order to flow naturally.

The questionnaires were pilot-tested on 10 randomly selected participants to ensure that they were comfortable when responding to questions. Face-to-face interviews were conducted by trained enumerators using a structured questionnaire on selected rural (*n* = 100), peri-urban (*n* = 100), and urban (*n* = 100) households. The participating households were selected based on ownership of village chickens and willingness to participate in the study. The enumerators and local leaders identified households. Enumerators were identified and recruited from the communities with assistance from the local leaders. At each site, 25 participants (aged 18 years or above) from different households were selected for an interview. The questionnaire encompassed aspects on household demography, flock sizes, reasons for keeping village chickens, and the household’s daily diet composition and diversity. In this context, flock size refers to the number of hens, roosters, pullets, and cockerels. The flock sizes were reported by the household member(s) and directly assessed by both the household member(s) and researchers.

### Dietary Diversity Assessment

2.4

Food items were classified into seven food groups according to the World Health Organization (WHO) and Infant and Young Child Feeding (IYCF) model chapter: the staple food group included four items (bread, rice, noodles, or other grains; white potato/yam, manioc, or other tuber; commercial baby cereal; and porridge), the animal source food group included five items (beef, pork, lamb, goat, rabbit, or deer; chicken, duck, or other birds; liver, kidney, heart, or other organs; eggs; and fish), the milk product group included two items (tinned/powder or fresh milk and baby formula), the green leafy/orange color vegetable group included two items (dark green leafy vegetables and pumpkin, carrots, squash, or sweet potato), the pulses group included one item (beans, peas, or lentils), the oils/fats group included one item (oils, fats, and butter), and the seeds group included one item (nuts).

Consumption of each food group was defined as “yes” when at least one food item within the food group had been consumed and “no” if no food item within the food group had been consumed. Dietary diversity was assed using a 24-hour dietary recall and a food variety score method. Participants were directly asked by a trained interviewer about the food items that they consumed in the last 24 h. The consumed food items were then recorded on a paper sheet. Food variety score is a proxy indicator used to measure nutrient adequacy [2]. The food variety score (FVS) range was from zero to nine. When the sum of the number of food items was less than nine, the sum was regarded as the score. When the sum of the number of food items was nine or more, nine was given as the score.

### Statistical Analyses

2.5

All the data were analyzed using SAS (2010). The PROC FREQ procedure for chi-square was used to compute the associations between distance from the city center with household demographics and village chicken flock sizes. The General Linear Model (GLM) was used to assess the ranked contributions of village chickens to dietary diversity and to calculate the mean scores of food variety. Regression analysis was used to determine the relationship between distance from the city center and flock sizes with the dietary diversity of households. The response surface regression (RSREG) procedure was used. The model used was as follows: $$\text{Y}=\beta_{0}+\beta_{1}\text{D}+\beta_{2}{\text{D}}^{2}+\text{E}$$ where Y is the response variable (dietary diversity); β_0_, β_1_, and β_2_ are regression coefficients; D is the distance from the city center; and E is the residual error.

## Results

3

### Household Demographics and Characteristics

3.1

There was an association between the rural–urban gradient and training on livestock production (*p* < 0.05) and village chicken flock size (*p* < 0.001). Over 17% of study participants had an educational level of grade 8–12 along the rural–urban gradients (Table 1). Less than 4% of participants across the gradients had received tertiary education. Less than 10% of participants received training on livestock production (*p* < 0.05).

### Distance from City Center and Village Chicken Contributions

3.2

Meat, followed by eggs and income derived from village chicken sales, was ranked as the top contributor to household dietary diversity across the rural–urban gradients (Table 2). A linear relationship was observed between distance from the city center and income. No quadratic relationships were detected. Income derived from village chicken sales increased with an increase in distance from the city center (*p* < 0.05). Village chicken offal and manure were ranked as the smallest contributors to household dietary diversity across the rural–urban gradients.

### Flock Sizes and Village Chicken Contributions

3.3

Meat, eggs, offal, income, and manure from village chickens contributed differently to household dietary diversity as the flock sizes changed (Table 3). The contribution of village chicken meat increased with an increase in flock size (*p* < 0.01). A quadratic relationship between village chicken flock sizes and meat contribution was observed (*p* < 0.01). The contribution of income derived from village chicken sales increased with an increase in flock size (*p* < 0.05). A quadratic relationship between the contribution of village chicken offal and flock size was observed (*p* < 0.01). The contribution of village chicken manure increased with an increase in flock size (*p* < 0.001). A quadratic relationship between the contribution of village chicken manure and flock size was observed (*p* < 0.001).

### Relationship between Distance from City Center and Dietary Diversity

3.4

The relationship between distance from the city center and households’ 24-hour dietary recalls is shown in Table 4. A quadratic relationship was observed between distance from the city center and consumption of LDFs (*p* < 0.05). There was a linear relationship between distance from the city center and the consumption of vegetables. Consumption of vegetables increased with an increase in distance from the city center (*p* < 0.01). A quadratic relationship was observed between distance from the city center and the consumption of pulses (*p* < 0.001). The relationship between distance from the city center and the consumption of staple foods, oil/fats, seeds, and milk products was not significant (*p* > 0.05).

### Relationship between Flock Sizes and Dietary Diversity

3.5

The relationship between village chicken flock sizes and households’ 24-hour dietary recalls is shown in Table 5. A linear relationship between village chicken flock sizes and the consumption of livestock-derived foods (LDFs) was observed. Consumption of LDFs increased with an increase in village chicken flock sizes (*p* < 0.05). There was a linear relationship between village chicken flock sizes and the consumption of vegetables. Consumption of vegetables increased with an increase in village chicken flock sizes (*p* < 0.01). The relationship between village chicken flock sizes and the consumption of staple foods, pulses, oil/fats, seeds, and milk products was not significant (*p* > 0.05).

### Distance from City Center, Flock Sizes, and Food Variety Scores

3.6

The relationships of distance from the city center with village chicken flock sizes and households’ FVSs are shown in Table 6. Linear and quadratic relationships were observed between the distance from the city center and village chicken flock sizes. Village chicken flock sizes increased with an increase in distance from the city center (*p* < 0.001). There was a linear relationship between distance from the city center and FVS. The FVS increased with an increase in distance from the city center (*p* < 0.05).

The FVS was affected by village chicken flock sizes (Figure 2). The food variety score was the highest (9.10 ± 0.29) in households that had a flock size of 10–14 chickens (*p* > 0.05). The FVS was the lowest (8.30 ± 0.34) in households that had a flock size of less than five chickens (*p* > 0.05).

## Discussion

4

The majority of the study participants were female, and most of them had no training on livestock production. These findings suggest that women are mostly responsible for village chicken management, as also reported by Getachew et al. [21]. Village chickens are the most kept livestock species and are mainly owned by women [22,23]. Furthermore, livestock farming activities are mainly carried out by women, especially village chicken husbandry, animal production, and livestock management [24].

The finding that village chicken meat and eggs are the main contributors to house-holds’ dietary diversity concurs with Ambikapathi et al. [25]. Eggs are produced through-out the year and they require less time and energy when cooking [20]. Keeping village chickens had a positive linear relationship with household dietary diversity [25]. Singh et al. [8] reported that larger village chicken flock sizes are positively related to increased egg production and consumption by households. Hence, this subsequently enhances the households’ dietary diversity and income [8]. Furthermore, food security is the main reason for keeping village chickens, followed by income generation [26].

The finding that income derived from village chicken sales is one of the main contributors to household dietary diversity concurs with Bellows et al. [27]. The primary purpose of keeping village chickens is for sales instead of household consumption [27]. As the flock size increases, the purpose of rearing village chickens changes from consumption to sales for income generation [28]. Village chickens are important in meeting households’ nutrition security by allowing to maintain balanced and nutritious diets and purchase other food items using income derived from their sales [29].

The quadratic relationship between distance from the city center and the consumption of LDFs could be influenced by household income and unstable flock sizes. High levels of small livestock mortalities, rearing livestock for sales, and reserving them for ceremonies influence the consumption of LDFs [30]. The finding that the consumption of vegetables increased with an increase in distance from the city center could be influenced by agricultural practices, mainly on crop production. These findings agree with Chagomoka et al. [31], who reported that vegetables are mainly produced in rural and peri-urban areas.

The study findings suggest that increasing the size of village chicken flocks improves the dietary diversity of households. The observed relationships between village chicken flock sizes and the consumption of vegetables and LDFs reveal a significant contribution of village chickens to the dietary diversity of households. The observed increase in the consumption of LDFs with village chicken flock sizes could be influenced by agricultural practices, mainly on livestock production. The observation attests that village chicken flock sizes contribute to dietary diversity through meat and eggs. Chagomoka et al. [31] reported that crop and livestock production was more prominent in rural and peri-urban areas compared to urban areas. The finding that the consumption of vegetables increased with an increase in village chicken flock sizes could be influenced by chicken manure produced as a fertilizer to grow crops. Application of chicken manure improves crop production through enrichment of soil fertility [23]. Ndiweni [32] reported that chicken manure is organic and a rich source of nitrogen suitable for crop production. The use of chicken manure to grow vegetables could be one of the strategies most commonly used to diversify diets in vulnerable households.

The finding that peri-urban and rural areas had larger village chicken flock sizes agrees with Mseleku et al. [22], who reported an increase in chicken numbers along the urban–rural gradient. Wong et al. [33] also highlighted that the majority of village chickens are reared in rural areas. Scavenging village chicken farming systems are mostly prominent in rural areas [33,34]. In contrast, De Bruyn et al. [35] reported that small poultry flock sizes are owned by rural and peri-urban households. Similar findings were reported by Musa [36], who revealed that village chickens reared in urban areas produce large numbers of eggs and chicks per hatch compared to those in peri-urban and rural areas.

The findings suggest that this could be due to different management systems implemented in these areas [36]. In rural and peri-urban areas, there is a high prevalence of diseases due to poor veterinary services, feed shortages, and increased financial needs within households, leading to sales and consumption of chickens [35,37]. This, therefore, hinders the expansion of village chicken flock sizes within the aforementioned areas. In addition, the lack of husbandry practices such as housing is one of the main constraints that hinders village chicken production [36].

Households situated in urban areas had small village chicken flock sizes and low FVSs. This suggests that these households heavily rely on minimal dietary diversity [38]. In urban areas, the shortage of land is the main hindrance to flock size expansion due to limited space to construct village chicken houses [39]. A study by Yemane et al. [39] revealed that flock sizes are associated with household size and farming experience. The study revealed that households with large flock sizes were prominent in rural and periurban areas, with higher FVSs. These results are in contrast to the finding of Vispute et al. [40], who reported a low FVS in rural areas. Food variety score could be influenced by several factors, including household income and educational level. Seasonality is one of the main factors influencing dietary diversity in rural areas [14].

### Strengths and weaknesses

4.1

Our research had some strengths and weaknesses. A strength of this research is that it was conducted in rural, peri-urban, and urban areas. Most of the researches related to village chickens were conducted in rural and some of the peri-urban areas. There is few, if any, research on village chickens conducted in urban areas. Weaknesses of this study include little variation in ethnicity, age, and gender of respondents.

## Conclusions

5

The study revealed a positive relationship between village chicken flock sizes and households’ dietary diversity. Village chicken flock sizes highly influenced households’ consumption of animal-derived foods and vegetables. Assessing the availability of village chickens across rural–urban gradients is a worthy opportunity to utilize in order to improve households’ dietary diversity and alleviate poverty. Expanding village chicken flock sizes could potentially enhance the dietary diversity of households along rural–urban gradients. This could be achieved through the availability of agricultural advisors and veterinarian services, mainly in rural and peri-urban areas. It can be concluded that expanding village flock sizes could enhance the dietary diversity of households.

## Figures and Tables

**Figure 1 F1:**
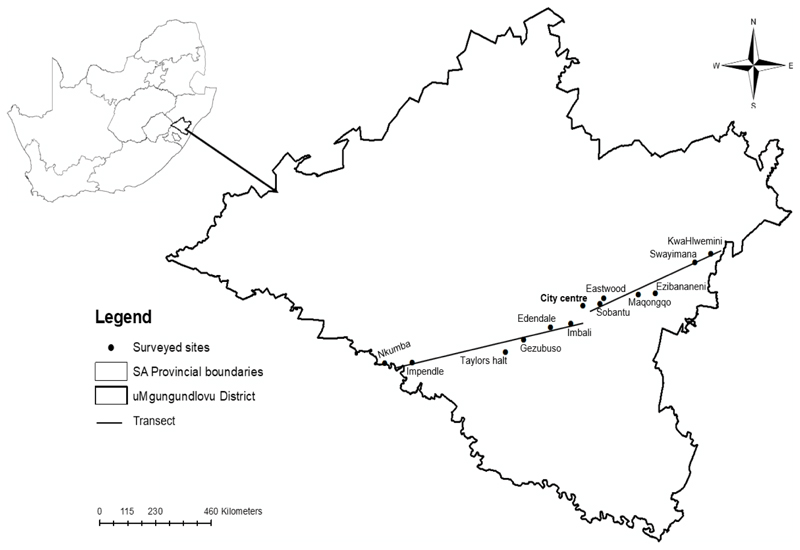
Location map of surveyed sites in Pietermaritzburg, uMgungundlovu District, KwaZulu-Natal, South Africa.

**Figure 2 F2:**
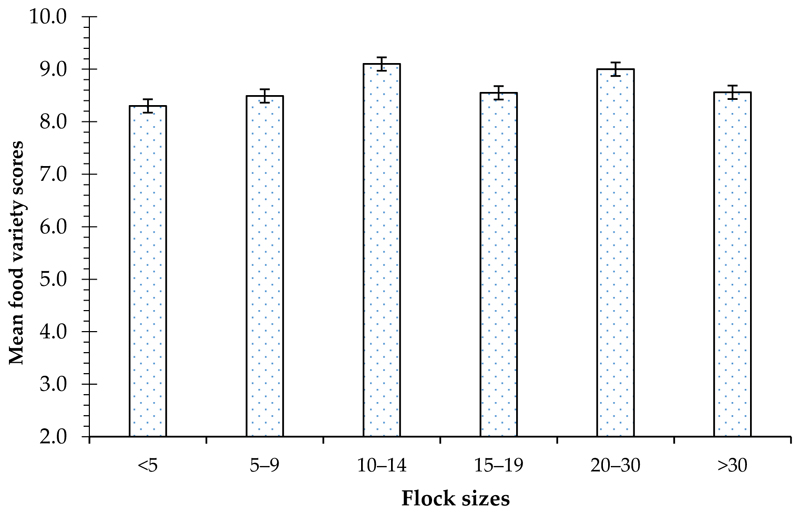
Mean food variety scores of households with different village chicken flock sizes..

**Table 1 T1:** Characteristics of participated households (%) along rural–urban gradients.

Characteristics	Urban	Site Peri-Urban	Rural	χ^2^	Significance
Gender					
Male	15.7	14.0	15.0		
Female	14.3	18.0	16.7	3.55	NS
Prefer not to say	3.0	1.3	2.0		
Age					
18–30	6.7	6.7	9.4		
31–50	16.2	13.1	11.5	6.37	NS
>50	9.8	13.5	13.1		
Marital status					
Single	18.1	16.4	20.8		
Married	11.6	10.9	7.9	13.08	NS
Divorced	1.4	0.3	0.7
Widowed	1.7	5.8	4.4		
Education level					
No formal education	2.9	3.9	2.5		
Grade 1–7	8.2	6.4	10.0	5.17	NS
Grade 8–12	20.3	19.6	17.4
Tertiary	2.9	2.1	3.9		
Household size					
1–5	20.6	15.0	16.4		
6–10	10.8	14.3	14.7	6.42	NS
>10	1.8	3.5	2.8		
Livestock training					
Yes	3.3	9.2	7.7	8.91	[Table-fn TFN1]
No	27.9	23.5	28.3	
Flock sizes					
<5	8.7	1.3	2.0		
5–9	4.7	3.7	6.0		
10–14	5.7	6.3	7.3	44.49	[Table-fn TFN2]
15–19	3.7	2.7	6.7
20–30	5.7	10.7	7.7		
>30	4.7	8.7	4.0		

**p* < 0.05;

****p* < 0.001; NS: not significant, *p* > 0.05. χ^2^–represents a chi-square value.

**Table 2 T2:** Distance from city center and village chicken contributions.

Contribution	Distance from City Center (km)			Regression Coefficients Linear
	<10	10–40	40–70	
Meat	1.43 ± 0.07	1.43 ± 0.08	1.35 ± 0.08	0.49 ^NS^
Eggs	2.31 ± 0.09	2.25 ± 0.09	2.19 ± 0.10	1.58 ^NS^
Offal	3.50 ± 0.14	3.36 ± 0.11	3.21 ± 0.11	0.29 ^NS^
Income	2.88 ± 0.15	2.65 ± 0.17	2.15 ± 0.17	1.88 [Table-fn TFN4]
Manure	3.31 ± 0.25	3.43 ± 0.20	3.57 ± 0.20	0.55 ^NS^

The lower the mean rank value is, the more important it is.

**p* < 0.05; ^NS^
*p* > 0.05.

**Table 3 T3:** Flock sizes and village chicken contributions.

Contribution	Flock Sizes						Regression Coefficients	
	<5	5–9	10–14	15–19	20–30	>30	Linear	Quadratic
Meat	1.75 ± 0.13	1.55 ± 0.12	1.42 ± 0.11	1.36 ± 0.13	1.35 ± 0.10	1.46 ± 0.11	1.20 [Table-fn TFN7]	−0.13 [Table-fn TFN7]
Eggs	2.53 ± 0.13	2.48 ± 0.11	2.12 ± 0.15	2.24 ± 0.13	2.14 ± 0.15	2.16 ± 0.16	0.31 ^NS^	−0.03 ^NS^
Offal	3.74 ± 0.21	3.53 ± 0.13	3.08 ± 0.15	3.04 ± 0.19	3.33 ± 0.19	3.53 ± 0.16	0.91 ^NS^	−0.13 [Table-fn TFN7]
Income	2.56 ± 0.19	2.11 ± 0.18	2.69 ± 0.33	3.00 ± 0.22	2.90 ± 0.26	2.33 ± 0.28	0.030 [Table-fn TFN6]	−0.02 ^NS^
Manure	3.53 ± 0.26	3.89 ± 0.24	3.71 ± 0.38	3.57 ± 0.27	2.94 ± 0.34	2.50 ± 0.41	1.86 [Table-fn TFN8]	−0.21 [Table-fn TFN8]

The lower the mean rank value is, the more important it is.

**p* < 0.05;

***p* < 0.01:

****p* < 0.001; ^NS^
*p* > 0.05.

**Table 4 T4:** Influence of distance from city center on households’ 24-hour dietary recalls.

Food Groups	Distance from City Center (km)				Regression Equations		R^2^	Significance	
	<10	10–40		40–70					
Staple foods	1.6 ± 0.07	1.6 ± 0.07		1.7 ± 0.07	y = −0.38x + 1.75 y = 0.13x^2^ – 0.38x + 1.75		0.022 0.006	NS NS	
LDFs	1.6 ± 0.09	2.0 ± 0.09		1.9 ± 0.09	y = 4.31x – 1.38 y = −1.06x^2^ + 4.31x − 1.38		0.004 0.276	NS [Table-fn TFN10]	
Vegetables	2.0 ± 0.12	2.1 ± 0.12		2.5 ± 0.11	y = 1.33x + 2.50 y = 0.46x^2^ + 1.33x + 2.50		0.222 0.051	[Table-fn TFN11] NS	
Pulses	1.4 ± 0.11	2.0 ± 0.11		1.6 ± 0.12	y = 4.27x – 2.13 y = −1.02x^2^ + 4.27x – 2.13		0.033 0.262	NS [Table-fn TFN12]	
Oil/fats	1.1 ± 0.02	1.0 ± 0.02		1.1 ± 0.02	y = −0.19x + 1.13 y = 0.06x^2^ − 0.19x + 1.13		0.066 0.018	NS NS	
Seeds	1.1 ± 0.08	1.3 ± 0.09		1.2 ± 0.08	y = −0.88x + 1.75		0.047	NS	
Milk products	1.2 ± 0.046		1.1 ± 0.05	1.1 ± 0.05	y = 0.25x^2^ − 0.88x + 1.75		0.052	NS	
					y = −1.31x + 2.38	0.011		NS	
					y = 0.31x^2^ − 1.31x + 2.38	0.073		NS	

The higher the mean value is, the more important it is.

**p* < 0.05;

***p* < 0.001;

****p* < 0.001; NS: not significant (*p* > 0.05). LDFs: livestock-derived foods.

**Table 5 T5:** Influence of village chicken flock sizes on households’ 24-hour dietary recalls.

Food Groups	<5	5-9	Flock Sizes		20–30	>30	Regression Equations	R^2^	Significance
			10–14	15–19					
Staple foods	1.5 ± 0.12	1.5 ± 0.11	1.8 ± 0.10	1.5 ± 0.12	1.7 ± 0.09	1.6 ± 0.10	y = 0.284x + 1.272 y = −0.039x^2^ + 0.284x + 1.272	0.002 0.012	NS NS
LDFs	1.6 ± 0.15	1.6 ± 0.15	1.6 ± 0.12	1.8 ± 0.14	1.9 ± 0.10	2.2 ± 0.12	y = 0.446x + 1.961 y = 0.102x^2^ + 0.446x + 1.272	0.240 0.081	[Table-fn TFN14] NS
Vegetables	1.9 ± 0.18	1.5 ± 0.17	1.8 ± 0.15	2.2 ± 0.18	2.0 ± 0.13	2.4 ± 0.16	y = 0.681x + 2.343 y = 0.144x^2^ + 0.681x + 2.343	0.175 0.084	[Table-fn TFN15] NS
Pulses	1.2 ± 0.08	1.0 ± 0.09	1.0 ± 0.07	1.3 ± 0.10	1.2 ± 0.06	1.1 ± 0.07	y = 0.031x + 0.951 y = −0.001x^2^ + 0.031x + 0.951	0.035 -	NS NS
Oil/fats	1.0 ± 0.04	1.1 ± 0.04	1.0 ± 0.03	1.0 ± 0.04	1.1 ± 0.03	1.0 ± 0.03	y = −0.141x + 0.887 y = −0.022x^2^ − 0.141x + 0.951	0.006 0.061	NS NS
Seeds	1.1 ± 0.05	1.0 ± 0.04	1.1 ± 0.03	1.0 ± 0.03	1.0 ± 0.03	1.0 ± 0.03	y = −0.215x + 1.422 y = 0.025x^2^ − 0.215x = 1.422	0.119 0.078	NS NS
Milk products	1.2 ± 0.15	1.1 ± 0.15	1.1 ± 0.12	1.3 ± 0.16	1.3 ± 0.11	1.1 ± 0.17	y = 0.346x + 1.826 y = 0.041x^2^ + 0.346x + 1.826	0.045 0.034	NS NS

The higher the mean value is, the more important it is.

**p* < 0.05;

***p* < 0.01; NS: not significant (*p* > 0.05).

**Table 6 T6:** Influence of distance from city center on flock sizes and food variety scores.

Measurement	Distance from City Center (km)			Regression Equations	R^2^	Significance
	<10	10–40	40–70			
Flock sizes	10.8 ± 0.96	20.3 ± 0.95	16.0 ± 0.95	y = 30.43x − 12.71 y = −6.96x^2^ + 30.43x − 12.71	0.043 0.103	[Table-fn TFN18] [Table-fn TFN18]
FVS	8.3 ± 0.24	8.7 ± 0.24	9.1 ± 0.24	y = 0.24x + 7.00 y = 0.05x^2^ + 0.24x + 7.00	0.021 -	[Table-fn TFN17] NS

The higher the mean value is, the more important it is.

**p* < 0.05;

****p* < 0.001; NS: not significant (*p* > 0.05).

## Data Availability

Data is contained within the article.

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
