# Peer review of "Relationship between Village Chicken Availability and Dietary Diversity along a Rural–Urban Gradient"

_nutrients, 2024, doi:10.3390/nu16132069_

Round 1

Reviewer 1 Report

Comments and Suggestions for Authors

Dear Authors,

Achieving food security globally and especially in developing countries is still a distant prospect. From this point of view, the research topic on household self-supply of poultry in South Africa should be considered relevant and important.

But the content of the manuscript is rather chaotic, and the issue posed in the title is one of many. In the Introduction, a lot of attention is paid to the dietary diversity of young children, which erroneously suggested that they would be the subject of the study. The authors write on L.53-54 about sustainable programmes, but in the next sentence they state that in LMICs the protein in the diet is mainly plant-based, which is what the sustainable consumption model assumes. There are also repetitions in this chapter, e.g. sentence L. 49-50 says the same thing as sentence L. 44-46 (nota bene there are authors' names instead of the reference number; also in L. 44), the sentence with reference no. 14 should be combined with the sentence from reference no. 11. The sentence in methodology L.91-92 should be in the introduction.

In the introduction, I would expect a characterisation of the current state of food insecurity in the country, statistical data, what is the pattern of food consumption and the average consumption of the different food groups, the energy structure of the diet, the spread of diseases, dietary deficiencies, etc. 

Regarding the methodology: why were the 12 localities selected (what do they have in common, any cut-off point?) and by what criterion was the selection made? What is the point of characterising sanitation and infrastructure and climate in this chapter (and in the manuscript in general)? These issues are unrelated to dietary diversity (so the content of L.92-103 is redundant). How were households selected for the study? What was the design of the questionnaire (ad L.125)? - redundant issues are raised in the discussion, not related to dietary diversity but rather to the way backyard chickens are reared. There should also be information in this section about the use of the 24-h recall Household Dietary Diversity Score method.

In all tables the unit should be annotated, where is the % of respondents, where is the average score, etc. Tables are numbered with one digit, in the text with two digits.   

As noted above, the discussion was dominated by issues of backyard chicken keeping. It should therefore be kept to a minimum and expanded to include dietary issues.

Comments on the Quality of English Language

There are several typos in the text, it is worth checking the use of plural and singular.  In L.62 – add ‘part’: as a PART of food system.

Author Response

The authors attached the responses to the comments.

Reviewer 2 Report

Comments and Suggestions for Authors

First and foremost, we thank the authors for the privilege of reviewing this manuscript. The manuscript delves into a crucial area of study, exploring the correlation between village chicken availability and dietary diversity of households along a rural-urban gradient in Pietermaritzburg, Msunduzi municipality in uMgungundlovu district, South Africa. We are eager to contribute to the enhancement of this manuscript, particularly in the “Materials and Methods section”, where there is a huge methodological concern:

- The authors should indicate the references supporting the established distances between urban, peri-urban, rural settlements and the city centre.

- The authors must clarify the type of discussions they applied to livestock association members and village headmen. For instance, was it a focus group or a consensus group?

- The authors should explain why they did not consider holding discussions with the women, as reported previously by the literature: “These findings suggest that women are mostly 231 responsible for village chicken management, as also reported by Getachew” (p. 9).

- The authors should clarify the period when the discussions were applied, where the discussion occurred and who conducted the discussions.

- The authors should clarify when the questionnaire was applied and how it was disseminated.

The authors must clarify the questionnaire's structure, questions, and scales. Reporting that “The questionnaire encompassed the same aspects as in the discussions” (p. 4) is an understatement.

For instance, were the flock sizes reported by household members or directly assessed by the researchers?

 -Could the authors elaborate on their insights from the discussion phase and how they applied them to construct the questionnaire? This information would greatly enhance the manuscript.

- For the “Dietary diversity assessment”:

i) Did the authors apply any food item adaptation from the Discussion phase?

ii) The authors must clarify the type of Dietary assessment used: 24-hour recall (as reported on p. 7)?

- The authors should apply the observing technique to the study's participants in their most natural setting—the village chickens in the different settlements—which would bring other insights and enable the triangulation of the questionnaire results.

Round 2

Reviewer 1 Report

Comments and Suggestions for Authors

Dear Authors,
I appreciate your commitment to improving the manuscript according to my suggestions. In its current version, it represents a good scientific study, has a much higher scientific value and is more interesting for readers.

Kind regards

Author Response

Thank you and we appreciate your comments.

Reviewer 2 Report

Comments and Suggestions for Authors

Dear authors,

Many thanks for considering the proposed comments and suggestions. Please reflect on the new comments and suggestions:

- The authors should clarify whether women's role in the FG discussion is related to local leaders or livestock association members.

- The authors should clarify and justify each FG's number and constitution.

- Could the authors elaborate on their insights from the discussion phase and how they applied them to construct the questionnaire? This information would greatly enhance the manuscript.

- The authors should clarify when i) the FG, ii) the questionnaire, and iii) the 24-hour dietary occurred.

- The authors should clarify better how was assed the 24-h dietary recall: was it administered by a trained interviewer or answered directly by the participant? Which type of support was used: e.g., a paper sheet?
